# Global surgery for medical students – is it meaningful? A mixed-method study

**Sofia Kühner** [1]*, **Solvig Ekblad**[2], **Jan Larsson**[3], **Jenny Löfgren**[4]

**1** Department of Medical Sciences, Uppsala University, Uppsala, Sweden, **2** Department of Learning, Informatics, Management and Ethics, Cultural Medicine, Karolinska Institutet, Stockholm, Sweden, **3** Department of Public Health and Caring Sciences, Uppsala University, Uppsala, Sweden, **4** Department of Molecular Medicine and Surgery, Karolinska Institutet, Stockholm, Sweden

* kuhner.sofia@gmail.com

## Abstract

### Introduction

There has been an increase in global health courses at medical universities in high-income countries. Their effect on students, however, is poorly understood. In 2016 an elective global surgery course was introduced for medical students at Karolinska Institutet in Sweden. The course includes a theoretical module in Sweden and a two-week clinical rotation in Uganda. The present study aimed to assess the format and determine its effect on students' knowledge of global surgery and approach towards patients of non-Swedish origin.

### Method

A mixed-methods design was used. Semi-structured case-based interviews were conducted individually with 18 students and analysed using qualitative content analysis. Examination scores and the course evaluation were analysed with Kruskal Wallis one-way analysis of variance, Pearson's Chi-square and a Wilcoxon signed-rank test as appropriate.

### Results

The course was appreciated and students reported gained insights and interest in global surgery. Students' ability to reason about global surgery issues was improved after the course. Students considered complicating aspects in the meeting with patients of non-Swedish origin. Students with abroad clinical experience felt less compelled to act on preconceptions.

### Discussion

The global surgery course at Karolinska Institutet is appreciated and students gained valuable knowledge. The case-based interviews acted as a catalyst for reflection and showed that students felt insecure as they lacked knowledge about globally common surgical conditions and struggled with generalized preconceptions of patients of non-Swedish origin. To further support students to integrate theoretical knowledge and professional development, we suggest the introduction of problem-based learning.

**Data Availability Statement:** Transcribed interviews (in Swedish) are stored at Karolinska Institutet, Department of Molecular Medicine and Surgery. The data set is not publicly available as student's informed consent to participate in the

interviews did not include an agreement to transcripts being published in full and due to a small sample size students can easily be identified which compromises confidentiality and violates the data protection policy of Karolinska Institutet. All the data that support the findings of this study are available on request (in Swedish) from registrator@ki.se pending ethical approval. Information provided in the manuscript and supporting information are sufficient to replicate the study, to analyze our data please request it as stated above.

**Funding:** The authors received no specific funding for this work.

**Competing interests:** We have read the journal's policy and the authors of this manuscript have the following competing interests: The second author is one of the founders and organizers of the global surgery elective course at Karolinska Institutet. This does not alter our adherence to PLOS ONE policies on sharing data and materials.

## Conclusion

The ability of the course to inspire students' commitment to global surgery is promising as this engagement is the key to reaching the goal of equitable health globally. Offering such courses is a step towards inspiring and recruiting the future clinicians and researchers needed for expanding the field of global surgery.

## Introduction

A global perspective is important when preparing medical students for their future profession [1–4]. The global burden of disease, travellers' medicine and immigrant health have been stated as key elements of global health (GH) education in medical school [5], but global health is more than that, being"the area of study, research and practice that places a priority on improving health and achieving equity in health for all people worldwide" [6]. To date, there is no consensus on how to teach students global health, and limited knowledge about the value and impact of existing courses [7–10]. A nation-wide study of final-year medical students in Sweden reported lack of knowledge and skills regarding global health [3]. Students' self-assessed knowledge corresponded to the amount of global health education they received during their medical education [3, 11]. In 2018, five of Sweden's seven medical schools offered elective courses in global health, and one integrated global health elements in other courses.

Surgery has previously been called the "neglected stepchild of global health" [12] but during the past decade global surgery has evolved as a field of its own. Growing evidence points to the vast need for surgical services globally and to their high level of cost-effectiveness [13]. There is therefore a call for education in global surgery. In Sweden, two specific courses in global surgery for medical students exist; one at the Karolinska Institutet (KI) and one at Lund University [10].

The KI course was introduced in 2016 and is offered biannually to fourth-year medical students. It is equivalent to two weeks of full-time studies. The course has two modules: the first is theoretical with lectures and seminars and is carried out in Sweden. The second part is a two-week clinical rotation at two hospitals in Uganda, with supervision and tutoring from both Swedish and Ugandan surgeons. The combination of theory and personal experience is thought to give the students' knowledge and personal insight into the field of global surgery and its implications abroad and in Sweden. A description of the curriculum, objectives and learning outcomes is given in **S1 Appendix**.

The present study sought to assess the Global Surgery course at Karolinska Institutets affect on students' development of knowledge and skills relating to global surgery, as well as exploring their approach to consultations with patients of non-Swedish origin in relation to the immigrant-health aspects of global health before and after the course.

## Methods and materials

The study used a mixed-methods design combining analysis of a formative assessment, results from the course evaluation and case-based interviews. Ethical approval was applied for from the Swedish Ethical Review Authority. The study was deemed exempt for ethical review as it was not needed for this type of study according to Swedish law (DNR: 2018/285-31/2).

The formative assessment consisted of a pre- and post-test, including eight objective response questions (ORQ) with three pre-set answer options and two subjective response

short-essay questions (SRQ), corresponding to learning outcomes regarding 'knowledge and understanding', see **S2 Appendix** for the translated formative assessment. The pre-test was distributed at the first course meeting and the post-test on the last day of the course in Uganda. The ORQ were developed to assess basic theoretical knowledge and the SEQ to assess understanding and ability to think critically and independently about global-surgery issues. The test was created by the course leaders, including the last author, and performed by students of the autumn semester of 2017 (n = 15) and spring semester of 2018 (n = 16). Maximum score was 16 points (ORQ = 10, SRQ = 6).

Anonymous course evaluations from the spring and autumn semesters of 2017 (n = 31) and the spring semester of 2018 (n = 16) were analysed. The evaluation consisted of standardized questions used for all courses at Karolinska Institutet and course-specific questions. Responses to the questions that remained constant over all semesters were analysed.

All students who had undertaken the course since 2016 and those participating during the spring semester of 2018 were invited to participate in the semi-structured interviews. The number of participants included in the study was based on information power [14]. Semi-structured, case-based interviews were conducted by the main author at Karolinska Institutet between March and May 2018. Each participant signed a written consent form before the interview.

The patient cases and the interview questions were developed by the second (SE) and fourth author (JL) to mimic a consultation with two patients of immigrant background at a Swedish health-care facility). SE is an experienced qualitative researcher and professor in Multicultural health and care. The interview questions were edited by the first (SK) and third (JLa) author after a test interview was performed, JLa also having great experience in semi-structured interview methodology. See **S3 Appendix** for a translated version of the case-based interview. SK conducted the interviews individually with each student, supervised by the fourth author (JL). A short introduction to the case was given and the students were then asked to reason clinically about the patient's symptoms and what questions they wanted to ask. They were accordingly provided with more information. When the student was content with the management, the case was ended with an epilogue describing all remaining information. The students were then asked questions about how they experienced the case and general questions about consultations with immigrant patients. Follow-up questions such as "Can you elaborate?" and "Can you give an example?" were posed to develop the answers further. Each interview lasted 45 minutes to 1.5 hours. The interviews were recorded and transcribed verbatim, resulting in 110 pages of single-spaced text.

## Analysis

Quantitative data were analysed using Microsoft Excel and MiniTab Express. For comparisons of proportions in the pre-test and post-test a paired *t* test was used for counts. A p-value $< 0.05$ was considered statistically significant. Ordinal data from the course evaluation were derived from a 5-point Likert scale.

Qualitative content analysis according to Malterud (1998) [15] was used to analyse the transcripts from the interviews and short-answer questions from the course evaluation. The material was investigated for primary themes and then condensed and shortened with the core message retained. The condensed text was explored for statements or sentences of relevance for the study aim. These constituted the meaning units and were further categorized under codes and sub-codes. Sub-codes were compared between exposed and non-exposed students. The material from each sub-group was summarized in an abstracted description that was defined by a headline and comprehensive quotes from the original material. In the last phase of re-contextualization, the complete material was analysed to see whether the descriptions of the subgroups still gave a satisfactory picture of the context in the original transcripts.

## Results

### Formative assessment

Of 31 students, 25 (80%) participated in both the pre- and post-test (maximum score = 16 points). The median score in the pre-test was 5 (31%, IQR: 3) and in the post-test 11 (69%, IQR: 3) (T-value = -8,48 p < 0.0001). All students improved in the SRQ, see Fig 1 for an overview.

### Course evaluation

Of the 47 course participants invited to evaluate the course, 38 (80.9%) responded. All students had a positive overall impression of the course, had gained useful skills and would recommend the course to a fellow student. The clinical training in Uganda was considered meaningful by all students. Detailed descriptive statistics from the standardized questions can be seen in **S4 Appendix**. The answers to the open-ended questions in the course evaluation were evaluated and resulted in two independent themes: educational gains and personal outcomes, and opinions about the course (Table 1).

All the students stated that they had gained practical skills applicable in Sweden. They described new knowledge of 'medical management with limited resources' such as how to use only history-taking and physical examination for making a diagnosis, how to work effectively with limited material and human resources.

'The importance of clinical presentation and patient history. The importance of resource management. The family's' potential role as part of mobilization strategies and care of the patient.' Student 1, spring 2018.

The students learned about new conditions and for the first time experienced late presentation of disease such as cancer. They described having learned to interpret signs of disease on black skin. The students discussed new insights into culture in relation to health and disease, and many described having a deeper understanding of patients from another country and culture in relation to health-seeking behaviour and disease.

'I've learned to understand people of other cultures more' Student 2, spring 2018.

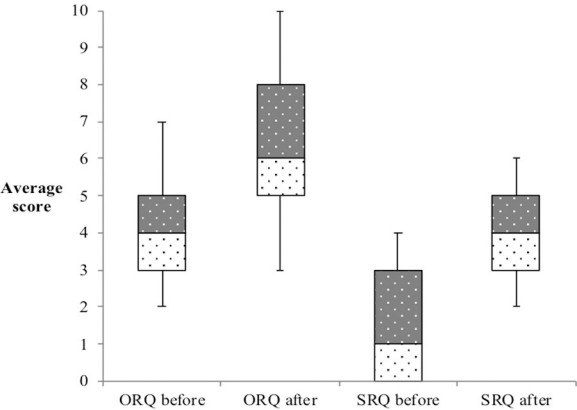

**Fig 1. Students formative test scores before and after the course.** Box plot showing students' (n = 25) test scores with first (white dotted) and third (grey dotted) quartile before and after the course for the ORQ and SRQ, the black line indicates median. Whiskers indicate min and max scores. Maximum score for the MCQ was ten (10) and for the SEQ six (6).

**Table 1. The coding tree of course evaluation short-answer questions.**

| Theme | Code | Sub-code |
|---|---|---|
| **Educational gains and personal outcomes** | Practical skills | Medical management with limited resources |
| | | Culture in relation to health and disease |
| | | New conditions and presentation of disease |
| | Reflection and comparison | Differences in health and healthcare systems globally and locally |
| | | Swedish considerations |
| | Future interest for global surgery | Future engagement in the global surgery field |
| | | Uncertain |
| **Students' opinions about the course** * | Clinical rotations in Uganda | Uganda and the different hospitals as locations |
| | | Educational environment |
| | | Preparation, information and communication |
| | | Request for longer rotations |
| | Theoretical part in Sweden | Unclear curricular aims and information about examination |
| | | Lectures |

* Results from 'Students' opinions about the course' can be seen in **S4 Appendix**.

Students described insight into and perspective on differences in health and healthcare systems globally and locally and how social and economic circumstances affect patients and the health care system.

'I've learned about external factors for health care and disease in a population' Student 8, spring 2017.

Students reflected on differences between Uganda and Sweden and often expressed gratitude for the Swedish resource-rich setting, the doctor-patient relationships and integrity. Some concluded that Swedish health care could be more resource-effective.

'Gratitude for Swedish health care resources and development, do not take this for granted as a patient or healthcare worker' Student 10, autumn 2017.

Almost all the students expressed that the course inspired them to future engagement in global surgery and global health at large. Forms of involvement mentioned were international clinical work on short or long assignments, research or policy development abroad or in Sweden, and future international student electives.

## Interviews

Eighteen students participated in the interviews, 11 who had completed the course (exposed), and seven who were currently participating in the theoretical module (non-exposed). Four of the exposed and three non-exposed had participated in a previous global health course. Our analysis resulted in two themes: patient-associated aspects believed to influence the management of patients of non-Swedish origin, and students' reactions and actions associated with the management of patients of non-Swedish origin (Table 2). The interview coding matrix can be seen in S1 Dataset.

**Theme I. Patient-associated aspects believed to influence the management of patients of non-Swedish origin.** *Code 1. Background of significance.* Initially students often stated

**Table 2. Coding tree of the semi-structured interviews.**

| Theme | Code | Sub-code |
|---|---|---|
| I. Patient-associated aspects believed to influence the management of patients of non-Swedish origin | I. Background of significance | Social determinants |
| | | Previous traumatic life experience |
| | | Taboo subjects |
| | II. Potential differences in medical culture | Previous experience of disease and health care |
| | | Expectations of investigations and treatment |
| | | Understanding of the Swedish health care system |
| | | Patient–doctor relationship |
| | III. Obstacles to communication between patient and health care | Language barrier |
| | | Level of education |
| | | Patients' health literacy |
| II. Students' reactions and action associated with the management of patients of non-Swedish origin | I. Problem solving | Consultation technique |
| | | Educate the patient |
| | | Handling external factors |
| | II. Medical management | Knowledge of aetiology and epidemiology |
| | | Reactions to the unknown |
| | III. Students' feelings and previous experience of influence | Reflecting on previous experience |
| | | Emotional reaction |
| | | Prejudice versus facts |
| | | Lessons learned |

that a patient's non-Swedish origin did not affect their views or attitudes, but later they described numerous situations where patients' origin could play an important role. Exposed students were in general quicker at considering the patients' social determinants of health as important for the case management. The students mainly discussed experienced or potential situations where background could be of negative influence. When reflecting on case number two (a woman subjected to female genital mutilation), many students took into consideration women's status and role in their home country and how that could influence how a female patient communicates. They mentioned a concern that she could be oppressed by her husband.

> 'I get the image of her husband being very authoritarian and that she doesn't have that much to say about anything, this makes her quiet and reserved.'—Non-exposed, student 6.

Students recognized that patients of non-Swedish origin could to a greater extent have had previous traumatic life experience and that it could influence the consultation. Trauma was a word used by students during the interviews, sometimes specified to war, famine, violence, migration and disease. These traumatizing events were, it was reasoned, affecting health-seeking behaviour, making patients seek health care early or too late. They also believed the patients may have issues they need help with, in addition to the symptom they are presenting with; or that the symptoms were a result of previous experiences.

> 'Always when you realize that someone is new to this country, I think: is this PTSD? Is there something that has happened in her home? There is generally a reason for leaving your country . . . if she has experienced horrible things?'–Exposed, student 4.

Students brought up possible 'taboo subjects' as being relevant to the second case. They believed there could be cultural differences in talking about sex, sexuality, genital organs and psychiatric illness.

> 'And also, a taboo. . . so it's clear that it affects how, depending on the culture, how much you talk about feelings and intimate subjects. If you're from a culture where you do less of this, it's obvious that it affects how you talk, how much you talk about it and when you start speaking more freely'—Non-exposed, student 7.

*Code II. Potential differences in medical culture.* Students acknowledged differences between countries regarding health-care systems and how health care professionals interact with patients. This could be interpreted as a type of "medical culture". All the students believed that patients' previous experience of disease and health care were of great influence. Experience of health care was believed to be linked to country of origin and that country's healthcare system. The students reasoned that experience of delay in receiving adequate health care in a country with few resources would negatively affect trust in the health-care system. Some stated that patients of non-Swedish origin more often had higher expectation of investigations and treatment and expected that the patient-doctor relationship would be more authoritarian and male-dominated.

> 'Then there is the cultural discrepancy with people from other cultures and countries expecting other things from the hospital, to feel confirmed and feel that something has been done, they need a prescription or antibiotics'–Exposed, student 2.

> 'I imagine or think that I as a young woman don't instil the same trust always, in everybody, but maybe this is more so in patient groups with another culture where it's more common that doctors are male and older and that they have a more authoritarian than we are used to'–Exposed, student 3.

Lack of understanding of the Swedish health-care system, with different levels of care, equal access for all, low fees, and doctor confidentiality, was stated by the students to be common in immigrant groups. This was expected to make such patients less able to seek help at the correct level, scared of potential costs and, for immigrants without documents, and fearful of being reported to authorities.

*Code III. Obstacles to communication between patient and the health care system.* When meeting patients of non-Swedish origin, students believed communication was the most difficult aspect, mainly due to possible language barriers. The students spoke about struggles and lack of experience in how to work with an interpreter. They worried about a low level of education in this patient group negatively affecting the patients' understanding and their medical reasoning. The students, however, mentioned 'health literacy' as less related to a patient's country of origin and acknowledged that large differences regarding health literacy also exist in the general Swedish population.

> 'Frustration, usually over not being able to communicate, not knowing if what I say is grasped. Usually, they say that they understand and maybe they can comprehend what I'm saying on a basic level but not the nuances. [. . .] It's more of monologue when there is a language barrier. I can't understand their feelings and they have difficulties expressing them.'–Non-exposed, student 1.

**Theme II. Students' reactions and action associated with managing patients of non-Swedish origin.** *Code I. Problem-solving.* Students often spontaneously reflected on ways to

overcome the difficulties mentioned above. Daring to ask about the patients' fears, thoughts and expectations was believed to help overcome some difficulties associated with the patient's background and possible previous experience. Asking these questions was described by many students as a way to avoid being judgmental. The questions could also improve understanding of the patient's background, avoiding conflict and increasing trust.

'In the meeting with people of other cultures it's important to be open-minded. There can be aspects you fail to understand and/or control that define the patient a lot. Asking is really the only thing you can do,'–Exposed, student 2.

Students felt they should take more time to 'educate the patient' about the Swedish health care system, concept of disease and treatment. They also wanted to make sure they would have extra time when an interpreter was needed.

*Code II. Medical management.* Students reasoned about differences in the panorama of diseases depending on country of origin or travel history and concluded that knowledge about aetiology and epidemiology is important. Many students also stressed that the common conditions remain the same in all patients, regardless of origin.

"I think I would have considered the diseases I know as my first priority. If it turned out not to be that or if there were some unusual symptoms that I didn't recognize I would have had some other diseases in mind, that are more uncommon "–Non-exposed, student 6.

In the case of the sick child, the students were still sure of the diagnosis and management, but when feeling the need to advise a senior colleague this was mainly due to conflict with the parents. In the case of the patient subjected to female genital mutilation (FGM), most students said they lacked knowledge of this and described feeling helpless when faced with a patient with an unfamiliar problem. They resolved to ask senior colleagues for help immediately, refer the patient or reschedule and read up on medical management.

*Code III. Feelings and experience.* Unawareness of conditions associated with immigrants' health, such as FGM, and also cultural traditions and ways, evoked an 'emotional reaction' and most students felt powerless and insecure. There was a fear of offending the patient or making an incorrect medical judgment resulting in substandard management, due to not considering differences in culture and disease enough, or considering them too much.

"It makes you insecure, for sure. Will I manage this correctly? Will I step on someone's toes? Who am I to tell you how to manage this in the best way? When I know nothing about it."–Exposed, student 3.

Frustration and sadness were expressed about global inequities in health. Students who had participated in an elective course abroad 'reflected on previous experience' and mentioned their clinical experience abroad as having given them a deeper understanding of possible difficulties when patients from other countries encounter Swedish health care.

"Of course, you realize, when you have experienced other types of health care system, how doctors treat patients and other things common in Uganda. Then you realize that there is a major shock when people come to Sweden."–Exposed, student 2.

Exposed students and students with non-Swedish parents, or who had work experience with immigrant patients, mentioned this as of value in realizing how much individuals'

experience can differ, regardless of country of origin. Students considered 'prejudice vs. facts', and felt conflicted about what was actual knowledge and what was prejudice regarding patients in relation to their background. Exposed students in general were reluctant to draw conclusions based on preconceived ideas; they reflected on the origin of this presumed knowledge. The non-exposed students who reflected on their own bias were more prone to act on assumptions based on a patient's background or origin. They were, as were the exposed students, aware of having prejudices but less negative about having them and some expressed them as beneficial, making them manage the case quicker and better. In general, students stressed the importance of having an open mind towards all patients and being aware of what you do not know, and what you presume you know.

> 'On the other hand, I can't go around suspecting things about people randomly. [. . .] I believe it's good to have some facts, something about epidemiology, and not only jump to conclusions based on one patient you have met before [from that country].'–Exposed, student 7.

> 'You don't want to but you do it subconsciously, treat people differently. [. . .] But, based on my own experience, I might discover things earlier and take them into consideration earlier' -Non-exposed, student 3.

Students expressed that the format of the interview was thought-provoking and made them reflect on new matters, such as the effects of previous experience and biases in general. Most felt that they had been able to reflect on possible problems in meetings with patients of non-Swedish origin and how to solve them.

> "That you get a case you just have to sit and talk about, it feels good, we should do this more and in this context. These cases feel typical, worried parents. There is so much more hidden in cultural aspects and experience, what am I missing? What passes us by without us noticing? It's remarkable, you need to take it to heart."—Exposed, student 9.

## Discussion

This study demonstrates that students' ability to analyse issues relating to global surgery and global health improves as a result of participating in the Global Surgery course at Karolinska Institutet. Participating students are satisfied with the course and acquired knowledge and skills relevant to their clinical work in Sweden. Furthermore, the study indicates that the course affected the students' attitudes towards and management of patients of non-Swedish origin.

The students expressed that participation in the course gave them a deeper understanding of how a patient's background can influence patients' views on health and healthcare. During the case-based interviews, students identified difficulties they associated with patients of non-Swedish origin, mainly communication barriers. They included both language barriers and use of an interpreter as well as the effects of educational level and health literacy. Communication barriers have been identified as important issues in several previous studies [16–19].

The students in this study expressed an awareness of, and struggled with, their assumptions about the non-Swedish patient, and they presented communicative tools they use to avoid acting on prejudice. These were in general universal and included asking all patients about fears, thoughts and expectations. This use of a generic strategy tallies with findings of physicians' strategies when trying to manage intercultural consultations [20]. Roth et al. call this

phenomenon "circling the undefined" [21] and raises concern that these communication strategies might not contribute to clarity. Generalisations about minority groups such as foreign-born patients can lead to differences in care and can thus be damaging. Although all students presented tools for combatting possible preconceptions, it is still important to address these aspects of bias in students' attitudes as part of their professional development and how the experience abroad can contribute to this. The students who had participated in the course were less willing to act on preconceptions related to the patient's origin and emphasised the individual's experience. This indicates a possible difference between students who had participated in an elective abroad and those who had not, illuminating the ongoing discussion regarding culturally appropriate care and the difference between developing cultural competence and patient-centredness [22].

There is doubt whether abroad electives genuinely bring a positive change in students' professional development, as some studies indicate a more prejudiced attitude afterwards [23–26]. Structured reflection has been pointed out as a key element in professional development in relation to overseas electives [23, 27, 28]. Reflections with the Swedish course leaders were scheduled once per week in Uganda to identify issues that needed to be addressed. These facilitated the learning process towards better understanding of disease epidemiology in different contexts, social determinants of health, resource constraints and the consequences for medical practice.

The case-based interviews acted as a catalyst for reflection and recollection of personal experience and shed light on prejudices and their effects. One-on-one or small-group discussions of patient cases after an overseas experience may be a way to facilitate reflection on how to make use of experiences from electives abroad. From the perspective of a medical university, it is important to address these issues as a part of all students' professional development and communication skills.

Previous research has found international experience positive for diagnostic skills, awareness of the importance of public health and preventive medicine, and interest in working internationally [1, 29–34]. There is however fear that elective students can do harm in the host country, mainly by disturbing the work of healthcare professionals but also through working outside their competence. The potential negative impact on the host country has not been investigated in the present course, however the students are always accompanied by Swedish teachers to mitigate this latter risk.

The overseas experience carried much value for the students and their professional development. We argue that international rotation is an important part of, and should be retained in, global health teaching; and that it should be available for all medical students. We further propose that global health aspects should be included in the curriculum for all future health care professionals as it can play an important role in improving care for all patients, including our large immigrant population. As the course reportedly inspired students to further engagement in global surgery, courses like the one studied have a potential to add value to the field of global surgery also in the long run as individual commitment ensures development of the field [2, 13, 35–37].

## Strengths and limitations

This study is to our knowledge the first use of a mixed-method approach in assessing global surgery education. The interviews are rich in information and information power is high based on the number of interviews. Trustworthiness in qualitative content analysis depends on three key elements: credibility, dependability and transferability [38]. As no data was excluded from the analysis and agreement was reached, credibility can be viewed as high. The

dependability is also considered high as data collection was limited to only two months, the interviews were semi-structured and no changes to the questions were necessary during the interviews. The interviewer is a female medical student from another university, we believe this similarity with the interview subjects contributes to the richness in the material. Fictional patient cases is used as a proxy for attitudes and behaviours in clinical practice as this avoids intruding on patients and students and is less likely to promote socially desired behaviours in the observed student. Therefore, for this study, fictional patient cases were appropriate. However, the applicability is limited compared to direct observations and future research may benefit from investigate student interaction with real patients.

With self-assessment, as in course-evaluations, there is a risk of systematically over- or underestimating skills and knowledge. The response rates for both the formative assessment and the course evaluation are high and largely congruous, limiting the risk of selection bias. Quantitative and qualitative study designs both carry certain weaknesses. The study designs are combined in order to minimize the effect of these weaknesses on the result and to give a more substantive picture of the course.

### Implications and future research

Global surgery courses at an undergraduate level can be beneficial for the individual student but also the global medical community at large. Overseas clinical rotations can be part of that education, preferably with high teacher presence and follow-up seminars. What is yet to be evaluated is the host's view of the course and how far it is bilaterally beneficial. We encourage education in global surgery, and urge all involved in global health courses to evaluate multiple aspects of their education to ensure quality and benefit for all parts.

### Conclusion

The format of the KI global surgery course with theoretical education in combination with an elective abroad was appreciated and beneficial to the students. To further support students in professional development and reaching curricular goals, we advocate for on-site reflection and follow-up seminars after abroad electives. The ability of this course to inspire future engagement in global surgery is promising, as the engagement of individuals is the key to reaching the global goal of equitable health.

### Supporting information

**S1 Appendix. The Global Surgery course at Karolinska Institutet curriculum.**
(DOCX)

**S2 Appendix. Translated formative assessment.**
(DOCX)

**S3 Appendix. Translated case-based interviews.**
(DOCX)

**S4 Appendix. Complementary results: Course evaluation.**
(DOCX)

**S1 Dataset. Interview coding matrix.**
(DOCX)

## Acknowledgments

This research was conducted as part of the main author's master's thesis at Uppsala university under the supervision of the fourth author. They wish to acknowledge the contributions of the other course leaders Joy Roy, Helen Sinabulya and Fredrik Lohmander, and all the staff at Mubende Regional Referral Hospital, Soroti Regional Referral Hospital and Mulago National Referral Hospital for their contribution to the global surgery course at KI, especially Dr Alphonsus Matovu and Sr Mary Margaret Ajiko. Tim Crosfield is also thanked for the language review.

## Author Contributions

**Conceptualization:** Sofia Kühner, Solvig Ekblad, Jan Larsson, Jenny Löfgren.

**Data curation:** Sofia Kühner.

**Formal analysis:** Sofia Kühner, Jan Larsson, Jenny Löfgren.

**Investigation:** Sofia Kühner, Jenny Löfgren.

**Methodology:** Sofia Kühner, Solvig Ekblad, Jan Larsson, Jenny Löfgren.

**Project administration:** Sofia Kühner.

**Resources:** Jenny Löfgren.

**Supervision:** Jan Larsson, Jenny Löfgren.

**Validation:** Jenny Löfgren.

**Visualization:** Sofia Kühner.

**Writing – original draft:** Sofia Kühner.

**Writing – review & editing:** Sofia Kühner, Solvig Ekblad, Jan Larsson, Jenny Löfgren.

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
