## [Decision Letter · Decision Letter 0]

8 Mar 2021

PONE-D-20-34713

Global surgery for medical students – is it meaningful? A mixed-method study

PLOS ONE

Dear Dr. Kühner,

Thank you for submitting your manuscript to PLOS ONE. We greatly appreciate your patience as your paper was reviewed. After careful consideration, we feel that it has merit but does not fully meet PLOS ONE’s publication criteria as it currently stands. Therefore, we invite you to submit a revised version of the manuscript that addresses the points raised during the review process.

The reviewers indicate several areas that need to be addressed. I also agree that the paper is too long. While it is important to include a complete description of the curriculum, I think it can be shortened and several items included as supplements. In particular, I would give consideration to providing tables 1 and 2, the appendix and the case descriptions as supplemental documents. Also, please also be sure to address the comment of reviewer #1 that no local staff or researchers were involved as co-authors in this study, nor were their contributions acknowledged in the paper. The study could not have been conducted without their assistance.

We look forward to receiving your revised manuscript.

Kind regards,

Richard Bruce Mink

Academic Editor

PLOS ONE

Journal Requirements:

2)  Please include additional information regarding the survey or and qualitative questionnaire used in the study and ensure that you have provided sufficient details that others could replicate the analyses. For instance, if you developed a questionnaire as part of this study and it is not under a copyright more restrictive than CC-BY, please include a copy, in both the original language and English, as Supporting Information, or include a citation if it has been published previously.

3) In the Methods, please discuss whether and how the questionnaire was validated and/or pre-tested. If these did not occur, please provide the rationale for not doing so.

4) Please clarify whether students consented to collection of their quantitative test scores for research purposes? Which type of consent was obtained - written or verbal?

5) We suggest you thoroughly copyedit your manuscript for language usage, spelling, and grammar. If you do not know anyone who can help you do this, you may wish to consider employing a professional scientific editing service.  

6) Thank you for stating the following in the Competing Interests section:

[We have read the journal's policy and the authors of this manuscript have the following

competing interests: The second author is one of the founders and organisers of the

global surgery elective course at Karolinska Institutet.].

7)  We note that you have indicated that data from this study are available upon request. PLOS only allows data to be available upon request if there are legal or ethical restrictions on sharing data publicly. For information on unacceptable data access restrictions, please see http://journals.plos.org/plosone/s/data-availability#loc-unacceptable-data-access-restrictions.

8) Please note that in order to use the direct billing option the corresponding author must be affiliated with the chosen institute. Please either amend your manuscript to change the affiliation or corresponding author, or email us at plosone@plos.org with a request to remove this option.

9)  Please include a copy of Table 5 which you refer to in your text on page 16.

10) We note you have included a table to which you do not refer in the text of your manuscript. Please ensure that you refer to Table 4 in your text; if accepted, production will need this reference to link the reader to the Table.

Reviewers' comments:

Reviewer's Responses to Questions

**Comments to the Author**

1. Is the manuscript technically sound, and do the data support the conclusions?

Reviewer #1: Yes

Reviewer #2: Partly

2. Has the statistical analysis been performed appropriately and rigorously? 

Reviewer #1: Yes

Reviewer #2: Yes

3. Have the authors made all data underlying the findings in their manuscript fully available?

Reviewer #1: Yes

Reviewer #2: No

4. Is the manuscript presented in an intelligible fashion and written in standard English?

Reviewer #1: Yes

Reviewer #2: No

5. Review Comments to the Author

Reviewer #1: The authors performed a mixed-methods study to assess the impact of an elective global surgery course, including theoretical modules and two-week rotations in Uganda, on students’ perceptions and knowledge of global surgical care delivery and attitudes towards patients of non-Swedish origin. The authors found that students with international experience reported being more comfortable meeting patients from non-Swedish origin, especially with regards to having fewer preconceptions and resulting potentially discriminatory actions. I applaud the authors for their extensive work but have some comments to improve their manuscript:

Major Comments:

1. Background, second paragraph: especially in global surgery, other areas of importance include “conscious resource utilization” and “understanding socioeconomic backgrounds/factors”. The authors may wish to discuss this briefly beyond a mere focus on disease burdens.

2. Background, course description: can the authors comment on the impact on local providers/trainees? They mention that Swedish teachers guide students, thus limiting the burden for Ugandan staff; however, it is worth mentioning whether these rotations displace local students/residents and/or have an impact on local staff’s autonomy.

3. Along those lines, this is worthy of further discussion in light of the obtained qualitative responses. For example, on Page 15, it is mentioned that “Ward rounds were in general hard to follow/difficult to hear. When it was bad it was a waste of time. When they were good, they were great! Unfortunately, they were more often bad.” Global health has long followed unbalanced power dynamics and been filled with unethical practices. While there has seemingly been much thought put into this when developing the course, it is critical to equally reflect on this during the course as well as upon reflection after the course. The authors are encouraged to recognize and reflect on this in their discussion beyond the brief sentence on Page 25 and postponing such reflection as “future research” on Page 28.

4. Similarly, on Pages 24-25 in the Discussion, it is reported that inequities were observed as “educational” by students and only led to a feeling of “gratitude towards the Swedish health care system.” It is important to recognize that global surgery courses have great potential but also are not intended to be “feel good trips” for students from high-income countries. This is worthy of discussion.

5. Can the authors discuss why no local staff or researchers were involved as co-authors in this study, nor acknowledged in the Acknowledgements, especially given their extensive administrative and teaching involvement in students’ rotations in Uganda?

Minor Comments:

1. Background, Page 6, “three-delay framework”: this should be clarified for the reader (i.e., delay in seeking care, reaching care, receiving care).

2. Methods, “MCQ questions”: “questions” is redundant, as the Q in MCQ stands for that.

3. Methods, “Only ten standard questions…”: can the authors discuss whether this resulted in any information bias, as 10 other standardized questions and 11 course-specific questions were excluded?

4. Abbreviations should only be introduced when used more than once after introduction. Abbreviations that are not used (i.e., GH) or used only once (i.e., LMIC) should be written in full only.

5. Page 22, “Students problematized”: this is best rephrased; “felt conflicted”?

Reviewer #2: This is a paper that provides a mixed methods analysis of a in person global health lecture series coupled with an on-site global health elective for medical students.

MAJOR

It is an ambitious paper, and quite possibly the longest paper I've read in medical education. That is my main concern, that they tried to do too much in this paper, without giving the reader the opportunity to really replicate anything that they are doing. The richness of this paper is not about changes in multiple-choice tests and how much people liked of course, but rather the qualitative reflections on culture differences experienced because of the rotation. Accordingly as someone who publishes quite often in global health education, I would suggest shifting the focus of this paper to that qualitative analysis alone. The course description could be included briefly in the background, but it is really this qualitative reflection that adds to the literature.

From a purely medical education standpoint, describing a course and saying that people liked it and had marginal changes in pre-and posttest scores, is quite low on the Kirkpatrick pyramid of educational outcomes. This can sometimes be overcome by actually sharing the curriculum and making it very easy for others to examine and replicate. For example, I am part of the team that created the global health education series available at sugarprep.org, and 1 of the ways we been able to publish so much on this work, is because even with lower outcomes of educational analysis, each time we are sharing exactly the curriculum and giving access to it for educators to use themselves. As written, this paper is a very long description that the authors created something and tested on it, yet beyond the brief description of the case discussions, there's nothing really applicable for another global health educator to use if they were trying to replicate this work.

This could be overcome, by either further describing and giving access to the curriculum, though that would only lengthen the paper. A better approach would be to relegate the course to the background, and dig deeper on the qualitative analysis which provides some addition to the literature.

Minor

There needs to be more description about the creation and validation of a multiple-choice test which itself has a robust methodology.

In the results, the authors share how many participated in the pre-and posttest, but do not provide the denominator here of how many participated in the course to be able to understand the response rate.

It's a little confusing that the pre-and posttest have different denominators of scores. This can be addressed by reporting the scores as a percent correct rather than a single digit number.

Additionally I'm not sure what is meant by the sentence there was a statistical improvement on group level. What does group level mean?

Did the scoring of the short essay questions include knowledge by the score of if it was a pre-or post response. This could certainly bias the results, as a score would want them to have improved.

More needs to be said about the 10 questions that were thrown out. With this decision made before scoring, or after.

The authors describe the rotation in Uganda as an exchange, yet this implies Ugandan students are coming to their program in Sweden. Exchange implies bidirectionality, yet none of what is described meets that definition. This is a clinical elective as described, not an exchange.

The authors appropriately reference the 3 delay framework, yet many readers will not know what this means. This should be expanded upon.

In the methods, the authors describe the results on the group level were compared before and after the course yet this sentence comes before describing that a pretest was conducted. This is a little confusing and they should be swapped.

6. PLOS authors have the option to publish the peer review history of their article (what does this mean?). If published, this will include your full peer review and any attached files.

Reviewer #1: No

Reviewer #2: No

---

## [Author Response · Author response to Decision Letter 0]

4 Jun 2021

Response to Reviewers

Dear editor,

Thank you for considering our manuscript for publication. We also thank the reviewers for their thorough revision and recommendations. We have updated the manuscript according to their comments and provide point-by-point responses below. Changes to the manuscript are shown using track-changes. 

Reviewer #1 

The authors performed a mixed-methods study to assess the impact of an elective global surgery course, including theoretical modules and two-week rotations in Uganda, on students’ perceptions and knowledge of global surgical care delivery and attitudes towards patients of non-Swedish origin. The authors found that students with international experience reported being more comfortable meeting patients from non-Swedish origin, especially with regards to having fewer preconceptions and resulting potentially discriminatory actions. I applaud the authors for their extensive work but have some comments to improve their manuscript:

Thank you for this positive comment! We provide our answers below. 

Major Comments:

1. Background, second paragraph: especially in global surgery, other areas of importance include “conscious resource utilization” and “understanding socioeconomic backgrounds/factors”. The authors may wish to discuss this briefly beyond a mere focus on disease burdens.

We totally agree that there are many aspects to global health education. This distinction has been excluded when shortening the manuscript. 

2. Background, course description: can the authors comment on the impact on local providers/trainees? They mention that Swedish teachers guide students, thus limiting the burden for Ugandan staff; however, it is worth mentioning whether these rotations displace local students/residents and/or have an impact on local staff’s autonomy.

The Swedish students are divided in small groups during the rotations. They participate in ward rounds, tuition and operations, together with Ugandan medical and/or nursing students and intern doctors and residents. The impact has not been investigated in the present study, but our impression is that both Swedish and Ugandan students and intern doctors/residents have appreciated the interactions and discussions. 

The Swedish teachers do not lead any of the ward rounds or teaching sessions in the hospitals. The role is rather to be on site to clarify issues and respond to questions from the Swedish students that are related to differences between Sweden and Uganda as well as logistical issues. When invited, the Swedish teachers have given lectures on specific topics. 

We have moved most of the information about the curriculum and course organisation to the S1 Appendix, please see our edited description on Ugandan staff. 

3. Along those lines, this is worthy of further discussion in light of the obtained qualitative responses. For example, on Page 15, it is mentioned that “Ward rounds were in general hard to follow/difficult to hear. When it was bad it was a waste of time. When they were good, they were great! Unfortunately, they were more often bad.” Global health has long followed unbalanced power dynamics and been filled with unethical practices. While there has seemingly been much thought put into this when developing the course, it is critical to equally reflect on this during the course as well as upon reflection after the course. The authors are encouraged to recognize and reflect on this in their discussion beyond the brief sentence on Page 25 and postponing such reflection as “future research” on Page 28.

Ward rounds in Sweden are very different from those that the students participate in in Uganda. At home in Sweden, the students are used to participate in ward rounds with 1-2 students together with senior colleagues. Patients often have their own room or two patients share the same room. In Uganda, there are many patients in very big wards. The space per patient is small and sometimes many students as well as intern doctors participate. Therefore, it is sometimes hard for the students to hear what is said. In addition, both doctors and patients have a tendency to speak in a low voice. The Ugandan English accent may also take a while to get used to for some students. 

While the students often complain about this, we still think that the ward rounds are an important part of the learning, not only the medical aspects but also that it helps them understand what it is like to be a student and a doctor in Uganda. It gives further understanding of the situation for the patients too. The students always bring this up for discussion at the scheduled reflection seminars. 

The theme “students’ opinion about the course” were moved to “Appendix x” to save space and also as this were not the main findings of the study. 

4. Similarly, on Pages 24-25 in the Discussion, it is reported that inequities were observed as “educational” by students and only led to a feeling of “gratitude towards the Swedish health care system.” It is important to recognize that global surgery courses have great potential but also are not intended to be “feel good trips” for students from high-income countries. This is worthy of discussion.

Thank you for this observation. We fully agree. It was not intended to come out that way. We discuss similarities and differences between Uganda and Sweden quite a lot, and the students reflect on these issues. We have never heard any student express themselves in a way that we have reason to believe that they see this as a “feel good trip” and these matters have been informally addressed before and during the elective. As these statements were a part of the course evaluation, we do not know what the students meant by this on a deeper level. We have elaborated on the need for reflection in the manuscript. 

5. Can the authors discuss why no local staff or researchers were involved as co-authors in this study, nor acknowledged in the Acknowledgements, especially given their extensive administrative and teaching involvement in students’ rotations in Uganda?

The study was carried out as a degree project for a Swedish medical student (SK) supervised by the fourth author (JK) and instructed by the third author (JL). SE participated in the study design, in particular the development of the patient cases and the interview guides. All of the work was carried out in Sweden. The findings were interesting and we decided to move write a manuscript for publication. The Ugandan teachers were not involved in the study design, data collection, data analysis and the writing of the manuscript. There are additional Swedish course leaders who are also not co-authors. Two of the main Ugandan partners are registered as PhD students at Karolinska Institutet with JL as main supervisor, however they did not participate much in the present study and therefore do not qualify to be co-authors according to the Vancouver criteria. 

We have added the names of the key partners in Uganda as well as the course leaders in Sweden in the acknowledgements. 

Minor Comments:

1. Background, Page 6, “three-delay framework”: this should be clarified for the reader (i.e., delay in seeking care, reaching care, receiving care).

Done.

2. Methods, “MCQ questions”: “questions” is redundant, as the Q in MCQ stands for that.

Edited. 

3. Methods, “Only ten standard questions…”: can the authors discuss whether this resulted in any information bias, as 10 other standardized questions and 11 course-specific questions were excluded?

We have clarified what was excluded in methods. 

4. Abbreviations should only be introduced when used more than once after introduction. Abbreviations that are not used (i.e., GH) or used only once (i.e., LMIC) should be written in full only.

This has been corrected.

5. Page 22, “Students problematized”: this is best rephrased; “felt conflicted”?

The paragraph has been edited.

 

Reviewer #2 

This is a paper that provides a mixed methods analysis of a in person global health lecture series coupled with an on-site global health elective for medical students.

MAJOR

It is an ambitious paper, and quite possibly the longest paper I've read in medical education. That is my main concern, that they tried to do too much in this paper, without giving the reader the opportunity to really replicate anything that they are doing. The richness of this paper is not about changes in multiple-choice tests and how much people liked of course, but rather the qualitative reflections on culture differences experienced because of the rotation. Accordingly as someone who publishes quite often in global health education, I would suggest shifting the focus of this paper to that qualitative analysis alone. The course description could be included briefly in the background, but it is really this qualitative reflection that adds to the literature.

Dear reviewer, thank you for this constructive advice. We have shortened the manuscript substantially and removed large parts of the results relating to the formative assessment as well as the course evaluation. Some of these data are presented in appendices for transparency purposes. 

From a purely medical education standpoint, describing a course and saying that people liked it and had marginal changes in pre-and posttest scores, is quite low on the Kirkpatrick pyramid of educational outcomes. This can sometimes be overcome by actually sharing the curriculum and making it very easy for others to examine and replicate. For example, I am part of the team that created the global health education series available at sugarprep.org, and 1 of the ways we been able to publish so much on this work, is because even with lower outcomes of educational analysis, each time we are sharing exactly the curriculum and giving access to it for educators to use themselves. As written, this paper is a very long description that the authors created something and tested on it, yet beyond the brief description of the case discussions, there's nothing really applicable for another global health educator to use if they were trying to replicate this work.

This could be overcome, by either further describing and giving access to the curriculum, though that would only lengthen the paper. A better approach would be to relegate the course to the background, and dig deeper on the qualitative analysis which provides some addition to the literature.

Thank you for this suggestion. We moved the full description of the course and complete curriculum as Appendix X. The cases have been translated and are available in Appendix X. 

Minor

There needs to be more description about the creation and validation of a multiple-choice test which itself has a robust methodology.

The questions were not strictly MCQ in format but objective response questions nonetheless and developed by experienced academic teachers based on the expected learning outcomes of the course. The test is only formative and is a way to appreciate to what extent the students learn what is intended with the course. We have edited the manuscript and use objective-response questions as descriptive phrasing instead of multiple-choice questions.

In the results, the authors share how many participated in the pre-and posttest, but do not provide the denominator here of how many participated in the course to be able to understand the response rate.

This information has been added to the manuscript. All teaching sessions in Sweden and in Uganda are mandatory. The pre- and post-tests have been administered at the first teaching session in Sweden and at the last day in Uganda. 

It's a little confusing that the pre-and posttest have different denominators of scores. This can be addressed by reporting the scores as a percent correct rather than a single digit number.

The denominators are presented in the updated manuscript. 

Additionally I'm not sure what is meant by the sentence there was a statistical improvement on group level. What does group level mean?

We have updated the manuscript to clarify this.

Did the scoring of the short essay questions include knowledge by the score of if it was a pre-or post response. This could certainly bias the results, as a score would want them to have improved.

The formative assessments were scored by both by JL and SK to ensure correct scoring. SK was not aware of which test was which. Each question only had one correct answer. 

More needs to be said about the 10 questions that were thrown out. With this decision made before scoring, or after.

The ten questions not included from the analysis of the course evaluation were questions that were not comparable over time as the phrasing or scoring changed. Only one question that was intact was excluded and it asked about whether or not schedules and information was handed out according to KI time-requirements. This question was not considered to be relevant for the study aim. We have updated the manuscript accordingly.

The authors describe the rotation in Uganda as an exchange, yet this implies Ugandan students are coming to their program in Sweden. Exchange implies bidirectionality, yet none of what is described meets that definition. This is a clinical elective as described, not an exchange.

This has been corrected. The Ugandan partners and teachers travel to Sweden, but not the students. 

The authors appropriately reference the 3 delay framework, yet many readers will not know what this means. This should be expanded upon.

This has been elaborated on but is now part of S1 Appendix with the curriculum. 

In the methods, the authors describe the results on the group level were compared before and after the course yet this sentence comes before describing that a pretest was conducted. This is a little confusing and they should be swapped.

This has been corrected. 

 

PLOS one editor comments

I would give consideration to providing tables 1 and 2, the appendix and the case descriptions as supplemental documents. 

This proposed change has been performed. 

Journal Requirements:

2) Please include additional information regarding the survey or and qualitative questionnaire used in the study and ensure that you have provided sufficient details that others could replicate the analyses. For instance, if you developed a questionnaire as part of this study and it is not under a copyright more restrictive than CC-BY, please include a copy, in both the original language and English, as Supporting Information, or include a citation if it has been published previously.

They have been added as supplementary files. 

3) In the Methods, please discuss whether and how the questionnaire was validated and/or pre-tested. If these did not occur, please provide the rationale for not doing so.

The formative assesment and the cases were developed by experienced teachers and interview questionnaire by qualitative researchers. The interview-questions were pretested on a student from Uppsala University, this has been updated in the methods. 

4) Please clarify whether students consented to collection of their quantitative test scores for research purposes? Which type of consent was obtained - written or verbal?

Ethical approval was applied for from the Ethical Review Board in Stockholm. The study was considered to be of minimal risk and ethical approval was therefore not needed. Students were verbally informed the results of the formative test would be used to improve the course. Course evaluations are accessible and published on the KI website as per routine. 

5) We suggest you thoroughly copyedit your manuscript for language usage, spelling, and grammar. If you do not know anyone who can help you do this, you may wish to consider employing a professional scientific editing service. 

We have employed Tim Crosfield, an independent translation and language editing professional.

6) Thank you for stating the following in the Competing Interests section:

We have updated the Competing Interests statement in your cover letter; we thank you for making the changes in the online submission form. 

7) We note that you have indicated that data from this study are available upon request. PLOS only allows data to be available upon request if there are legal or ethical restrictions on sharing data publicly. 

Please see updated Data availability statement in the updated cover letter. We thank you for updating the data availability statement on our behalf. 

8) Please note that in order to use the direct billing option the corresponding author must be affiliated with the chosen institute. Please either amend your manuscript to change the affiliation or corresponding author, or email us at plosone@plos.org with a request to remove this option.

We have tried to contact you in this matter without response. Please remove the direct billing option. 

9) Please include a copy of Table 5 which you refer to in your text on page 16.

This table has been renamed and moved. 

10) We note you have included a table to which you do not refer in the text of your manuscript. Please ensure that you refer to Table 4 in your text; if accepted, production will need this reference to link the reader to the Table.

This table reference has been edited.

---

## [Decision Letter · Decision Letter 1]

13 Jul 2021

PONE-D-20-34713R1

Global surgery for medical students – is it meaningful? A mixed-method study

PLOS ONE

Dear Dr. Kühner,

Thank you for submitting your manuscript to PLOS ONE. After careful consideration, we feel that it has merit but does not fully meet PLOS ONE’s publication criteria as it currently stands. Therefore, we invite you to submit a revised version of the manuscript that addresses the points raised during the review process.

As indicated by Reviewer #1, please include weaknesses in your limitations paragraph in the discussion.

We look forward to receiving your revised manuscript.

Kind regards,

Richard Bruce Mink

Academic Editor

PLOS ONE

Journal Requirements:

Reviewers' comments:

Reviewer's Responses to Questions

**Comments to the Author**

1. If the authors have adequately addressed your comments raised in a previous round of review and you feel that this manuscript is now acceptable for publication, you may indicate that here to bypass the “Comments to the Author” section, enter your conflict of interest statement in the “Confidential to Editor” section, and submit your "Accept" recommendation.

Reviewer #1: All comments have been addressed

Reviewer #2: All comments have been addressed

2. Is the manuscript technically sound, and do the data support the conclusions?

Reviewer #1: Yes

Reviewer #2: Yes

3. Has the statistical analysis been performed appropriately and rigorously? 

Reviewer #1: Yes

Reviewer #2: Yes

4. Have the authors made all data underlying the findings in their manuscript fully available?

Reviewer #1: Yes

Reviewer #2: Yes

5. Is the manuscript presented in an intelligible fashion and written in standard English?

Reviewer #1: Yes

Reviewer #2: Yes

6. Review Comments to the Author

Reviewer #1: The authors have adequately addressed most editor and reviewer comments. Other than reconsidering their limitations section (which currently only contains strengths, yet no study is flawless), I have no further comments and applaud the authors for their work.

Reviewer #2: The authors have nicely dressed the reviewer comments. It still remains that this is not a terribly actionable paper, yet the criteria for publication in this journal have been met.

7. PLOS authors have the option to publish the peer review history of their article (what does this mean?). If published, this will include your full peer review and any attached files.

Reviewer #1: No

Reviewer #2: No

---

## [Author Response · Author response to Decision Letter 1]

25 Aug 2021

Dear editor,

Thank you for considering our manuscript for publication. We also thank the reviewers for their thorough revision and recommendations the first and second time. We have updated the manuscript according to their comments and provide responses below.

Review Comments to the Author

Reviewer #1: The authors have adequately addressed most editor and reviewer comments. Other than reconsidering their limitations section (which currently only contains strengths, yet no study is flawless), I have no further comments and applaud the authors for their work.

Response: 

We thank you for this remark and opportunity to reflect on our work. We have updated the strengths and limitations section. 

Reviewer #2: The authors have nicely dressed the reviewer comments. It still remains that this is not a terribly actionable paper, yet the criteria for publication in this journal have been met.

Response: 

We are grateful for your comments and push towards a more generous article. 

Journal Requirements:

Response: 

All articles are still in print and no changes were made to the reference list.

---

## [Editor Report · Decision Letter 2]

31 Aug 2021

Global surgery for medical students – is it meaningful? A mixed-method study

PONE-D-20-34713R2

Dear Dr. Kühner,

We’re pleased to inform you that your manuscript has been judged scientifically suitable for publication and will be formally accepted for publication once it meets all outstanding technical requirements.

Kind regards,

Richard Bruce Mink

Academic Editor

PLOS ONE
---

## [Editor Report · Acceptance letter]

29 Sep 2021

PONE-D-20-34713R2 

Global surgery for medical students – is it meaningful? A mixed-method study 

Dear Dr. Kühner:

I'm pleased to inform you that your manuscript has been deemed suitable for publication in PLOS ONE. Congratulations! Your manuscript is now with our production department. 

Kind regards, 

on behalf of

Dr. Richard Bruce Mink 

Academic Editor

PLOS ONE